# Will the Real Immunogens Please Stand Up: Exploiting the Immunogenic Potential of Cryptococcal Cell Antigens in Fungal Vaccine Development

**DOI:** 10.3390/jof10120840

**Published:** 2024-12-04

**Authors:** Samantha L. Avina, Siddhi Pawar, Amariliz Rivera, Chaoyang Xue

**Affiliations:** 1Graduate School of Biomedical Sciences, Rutgers University, Newark, NJ 07103, USA; sa1357@njms.rutgers.edu (S.L.A.); ssp198@gsbs.rutgers.edu (S.P.); riveraam@njms.rutgers.edu (A.R.); 2Public Health Research Institute, New Jersey Medical School, Rutgers University, Newark, NJ 07103, USA; 3Department of Pediatrics and Center for Immunity and Inflammation, New Jersey Medical School, Rutgers University, Newark, NJ 07103, USA; 4Department of Microbiology, Biochemistry and Molecular Genetics, New Jersey Medical School, Rutgers University, Newark, NJ 07103, USA

**Keywords:** *Cryptococcus neoformans*, antigen, fungal vaccine, immune response

## Abstract

*Cryptococcus neoformans* is an opportunistic fungal pathogen that is a continuous global health concern, especially for immunocompromised populations. The World Health Organization recognized *C. neoformans* as one of four critical fungal pathogens, thus emphasizing the need for increased research efforts and clinical resource expansion. Currently, there are no fungal vaccines available for clinical use. Exciting new findings in cryptococcal vaccine development have identified whole cell-based and subunit-based vaccinations to help mitigate health risks and make commercialization attainable. Importantly, recent work has focused on how different cryptococcal cell-wall antigens modified in these vaccine candidates allow us to manipulate their immunogenicity to produce a desired long-term protective anti-fungal immune response. In this review, we discuss the different cryptococcal cell immunogens, namely the polysaccharide capsule, glucans, chitin/chitosan, mannoproteins, and extracellular vesicles, and their role in novel cryptococcal vaccination approaches. Additionally, we examine the immunological mechanisms responsible for protection in these vaccine candidates and the similar host response-stimulation pathways induced through different immunogen exposure.

## 1. Introduction

Invasive fungal infections (IFIs) are a growing global health concern as rates of fungal infections rise while fungal drug and vaccine development stagnate. It is estimated that approximately 1.6 million people die from IFIs worldwide annually; the number of cases is expected to rise as risk factors like climate change, immunocompromised population, and anti-fungal drug resistance increase [1]. Due to the concern of increased IFIs and associated co-morbidities, the World Health Organization organized the first Fungal Priority Pathogens List in 2022 to address research needs of prevalent fungal pathogens [2]. Listed at the top of the critical priority group is *Cryptococcus neoformans*. *C. neoformans* is an encapsulated yeast that is the causative agent of cryptococcal meningoencephalitis and the fungal pathogen responsible for the largest percentage of fungal meningitis cases worldwide. Recently, *Cryptococcus* species complex has been divided into seven different species [3]. To simplify the writing, here we will use traditional *C. neoformans* and *C. gattii* to describe the species complex. While *C. neoformans* is responsible for ~19% of HIV/AIDS-related deaths annually and immunocompromised populations are at the highest risk, the sibling species *C. gattii* can also infect immunocompetent hosts [4]. Thus, *C. gattii* is a high-risk primary pathogen. A key point highlighted on the fungal pathogen priority list is the lack of a vaccine in the global anti-fungal arsenal.

As highlighted by Rivera and colleagues 2022 review, the rationale of slow fungal vaccine development is due to a complexity of issues including socioeconomic considerations, similarity of fungal and mammalian cellular machinery, lack of fungal immunological understanding, and difficulty in mass commercialization of fungal vaccine research for limited populations [5]. Particularly, the targeting of fungal antigens that may induce a desired protective immune response while minimizing off-target effects in human hosts remains to be carefully defined and elucidated. The polysaccharide encapsulation of the *Cryptococcus* cell is unique in its anti-phagocytic properties and is considered a major immunogen masking component that contributes to immune evasion and ultimate dissemination throughout its host. Establishing methods to inhibit cryptococcal dissemination either via prophylaxis treatment or vaccination approaches prior to infection has proven difficult and is an area of active research in the field. In this review, we focus on highlighting known cryptococcal fungal cell components (namely the capsule, α/β-glucans, chitin/chitosan, mannoproteins, and extracellular vesicles) and how the immunogenicity of these antigens, or lack thereof, can be harnessed to shape and modulate the host immune response in novel cryptococcal vaccine approaches.

## 2. Cryptococcal Capsule

Pathogenic microorganisms containing a polysaccharide capsule have predominantly included bacteria (i.e., *Streptococcus pneumoniae*, *Haemophilus influenzae*, and *Neisseria meningitis*) [6] but are present on some fungi, most notably *Cryptococcus*. The *Cryptococcus* species complex has been historically well defined and identifiable by its unique encapsulation by a polysaccharide capsule that serves as a key virulence factor for survival in the host. The polysaccharide capsule of *C. neoformans* is connected to the cell wall as capsular polysaccharide or in a shed form identified as extracellular polysaccharide. These polysaccharides predominantly comprise glucuronoxylomannan (GXM) and galactoxylomannan (GalXM) at ratios of 90% to 10%, respectively (Figure 1) [7]. GXM is structured by a mannan backbone with xylose and glucuronic acid substitutions, while GalXM possesses a galactan backbone with mannose and galactose sidechain substitutions, which can further be substituted with xylose and glucuronic acid residues [8,9]. In clinical settings, cryptococcal meningitis is often diagnosed by the presence of cryptococcal capsule shedding in the spinal fluid or serum of patients [10,11]. The ability of the capsule to be continuously shed throughout the *Cryptococcus* cell lifespan and alter its size based on its environment greatly contributes to *Cryptococcus* survival and hijacking of host effector cells to migrate toward the central nervous system [12,13,14,15,16,17].

Understanding the structural basis of the capsule is key to further elucidating how we can modulate immunogenicity of the *Cryptococcus* cell. Namely, the GXM mannose backbone structural differences of capsule polysaccharide makeup have been utilized to classify serotypes of cryptococcal strains (serotypes A, B, C, D, and AD) based on their antigenic differences [18]. In the past few decades, studies have shown that highly immunogenic mannoproteins (MPs) constitute as a small fraction of the capsule makeup (~1–2% by mass) and reside in spatially different regions of the capsule [18]. MPs have not been shown experimentally to be covalently bound to the cryptococcal cell wall potentially due in part to their ability to be secreted outside of the cell and possession of Glycophosphoinositol (GPI) anchors to keep them bound to the cell wall [18]. Recently, one predicted mannoprotein, Krp1, in *C. gattii* was found to contribute to the capsule structure, as well as GXM shedding into the supernatant [19]. Whether there is a physical interaction between Krp1 and capsule remains to be determined. Additionally, loss of Krp1 resulted in diminished beta glucan synthesis. However, whether other mannoproteins contribute to altering capsule structure or synthesis remains to be seen and is discussed later in this review.

Studies on capsule synthesis and the modification of the capsule structure in nutrient-limited host conditions have been key to elucidating the role of capsule in *Cryptococcus* virulence and insightful in the progression of potential targets for vaccine development. Early studies by Kwon-Chung and Yang identified key capsule synthesis genes required for capsule synthesis (i.e., *CAP59*, *CAP60*, *CAP64*, and *CAP10*). Loss of any of these single genes results in acapsular strains of *Cryptococcus* [7,20,21,22,23,24]. Defects in capsule synthesis will result in attenuation of virulence and loss of immune evasion from responding phagocytes [25]. Cryptococcal vaccine studies focused on the role of the capsule on a glycolipid sterylglucosidase-deficient strain (*sgl1Δ*) showed that sterylglucoside accumulation in the *sgl1Δ* mutant alters the structural and physical properties of GXM. Consequentially, capsular GXM was required for complete host protection, as the acapsular *cap59Δ sgl1Δ* strain failed to induce a T_H_1 protective inflammatory response. Similarly, the cryptococcal vaccine candidate (Znf2^OE^) that overexpresses the zinc finger transcription factor Znf2 also requires the capsule for vaccine protection [26]. This study utilized an acapsular strain deficient in both GalXM and GXM in the Znf2^OE^ background. While the exact impact of Znf2 overexpression on GXM or GalXM was not defined, it was observed that sera from mice immunized with heat-killed Znf2^OE^ contained IgG and IgM antibodies that bound themselves to antigens highly abundant in the center of the Znf2^OE^ capsule, as compared to H99. Studies with various whole-cell vaccine candidates that affect different molecular pathways, and subsequent cryptococcal cell antigens, also suggest that these mutants require the capsule to act as the carrier or as a prop to expose their immunogens and thus induce protective immunological responses [26,27]. However, the observations that there is no protection with *cap59Δ* vaccination strategies and that acapsular strains are rapidly cleared in the host upon challenge suggest that exposure of immunogens in the cell wall alone are not sufficient for eliciting a protective immune response against cryptococcal infection.

The *Cryptococcus* capsule is known to be anti-phagocytic and to elicit T cell independent T_H_2 responses. The inability of GXM to induce memory T-cell activation by the host or induce antibody affinity maturation and immunoglobulin class switching contributes to the poor immunogenic potential of GXM [28,29]. Nevertheless, antibodies against GalXM and GXM capsular components have been explored for fungal vaccine and therapeutic treatment uses. Casadevall and colleagues generated monoclonal antibody 18B7 as a neutralizing antibody for cryptococcosis treatment and showed its ability to bind to four serotypes of *C. neoformans* with specificity to GXM in lung tissue retrieved from murine infection models [30]. Phase 1 human clinical trials were conducted for monoclonal antibody 18B7 as a therapeutic treatment for cryptococcosis; however, treatment only showed moderate amelioration of symptoms [31]. GXM-based vaccine approaches have focused on conjugating GXM to carrier proteins such as tetanus toxoid (GXM-TT), which induces antibody mediated protection and showed prolonged survival in vaccinated mice after *C. neoformans* challenge [32,33]. Another GXM-based vaccine is the GXM mimicking peptide, P13 conjugated to tetanus toxoid (P13-TT) [34]. In human immunoglobulin transgenic mice P13-TT immunization studies, subcutaneously vaccinated cohorts showed prolonged survival compared to controls [34]. This P13-TT protection was antibody-mediated, as vaccinated mice produced IgG2 and IgG4 antibodies against P13-TT and immunogenic idiotype-positive antibodies to GXM. Recently, semisynthetic glycoconjugate vaccines containing an identical synthetic decasaccharide M2 motif antigen bound to anthrax and CRM197 were found to induce protective antibodies against GXM but showed modest protection in murine models compared to controls [35].

Another antibody-based vaccine candidate is based on the predominant component of the capsule, namely GalXM, conjugated to antigenic carrier bovine serum albumin (GalXM-BSA) [36]. In GalXM-BSA vaccine studies, the BSA-conjugated GalGXM complex was able to induce passive IgM and IgG antibodies but saw no induction of host defense against *Cryptococcus* infection between vaccinated and unvaccinated cohorts [36]. Modifications need to be implemented to prolong protection and induce a CD4^+^ T cell-protective immunity. Cryptococcal capsule-based vaccine candidates are an active area of research. A better understanding of how capsule synthesis is regulated and altered while in different host conditions is important to move capsule-conjugated vaccine research forward. Moreover, identifying epitopes or synthetic modifications modeled off the capsule may chart a path forward for multivalent subunit-based pan-fungal vaccines.

## 3. Cryptococcal Glucans

The fungal cell wall is a critical surface structure that maintains cell integrity against biological, physical, and chemical stressors and decides the fate of the pathogen [37]. This rigid structure is also dynamic and flexible in nature to undergo morphologic changes during mating, budding, or cellular interactions, including the one with host cells [38].

The cryptococcal cell wall consists of glucans, chitin, chitosan, glycoproteins, melanin, and lipids [39]. These components are major fungal pathogen-associated molecular patterns (PAMPs) and help fungi to sense their surroundings and contribute to survival inside the host. The synthesis of precursors of cell-wall glucans involves the coordinated action of glycosyltransferase with donor sugar molecules, enzyme activities, and the availability of acceptor substrates [40]. Unlike *Saccharomyces cerevisiae*, *C. neoformans* has abundant α-glucans and chitosan polymers, followed by more β-1,6 linkages with minor levels of β-1,3 linkages [40,41,42].

The cryptococcal cell wall primarily consists of α-1,3-glucan linkages derived from membrane-bound α-glucan synthase, Ags1p, primarily associated with the outer cell wall [41]. Loss of *AGS1* results in loss of α-1,3-glucan and capsule, followed by the redistribution of β-glucans and chitin, making the cells more fragile [43]. These findings show support for the association of α-1,3-glucan in the binding of capsular polysaccharides. Although *C. neoformans* has a significantly lower percentage of β-1,3-glucan, the gene *FKS1* encoding for β-1,3 glucan synthase is essential [42]. The phenotypic defects observed in the loss of *FKS1* supports the idea that β-1,3 glucan plays a critical role in cell viability and capsule organization. Inhibitors of β-1,3-glucan synthesis (i.e., Echinocandins) have no effect on cryptococcal β-1,3-glucan production, possibly suggesting the presence of transporters that may pump out the compounds [42,44]. In comparison, the mechanism of synthesis for β-1,6-glucan is complex since no synthase enzyme has been identified. The synthesis depends on multiple genes, with a dominant role played by *KRE5* [45]. While there are seven KRE genes in *C. neoformans*, deletion of KRE5 alone or KRE6 with *SKN1* led to complete loss of β-1,6-glucan from the cell wall, resulting in compromised cell integrity, rendering it avirulent in an animal inhalation model of infection, as the yeast cells were unable to survive at host temperature [45].

Cell-wall glucans not only play an important role in cell integrity but also induce immunomodulatory effects in the host. Basso et al. evaluated the immunostimulatory activity of β-1,3-glucan-containing exopolysaccharide (EPS) isolated from the edible mushroom *Auricularia auricula* to phagocytes and to mice infected with *C. neoformans* [46]. Treatment with EPS resulted in the activation of innate cells like macrophages and dendritic cells after engagement of Dectin-1 receptor culminating in pro-inflammatory cytokine production and cell maturation via Syk-dependent pathway signaling. EPS treatment resulted in the upregulation of genes associated with host protection against *C. neoformans* and Dectin-1-mediated signaling in macrophages and enhanced the survival of *C. neoformans*-infected mice [46].

Glucan particles (GPs) can be used for dual purpose as a combined delivery system and an adjuvant for cryptococcal vaccine due to its ability to elicit protective immune responses. Glucan particles are recognized by complement and Dectin-1 receptors present on innate immune cells [47]. Mice immunized with GPs containing trapped ovalbumin resulted in T_H_1/T_H_17 CD4^+^ T-cell responses, followed by robust antigen-specific antibody responses [48,49]. Based on these protective responses, Specht et al. recombinantly expressed six purified cryptococcal proteins (Cda1, Cda2, Cda3, Fpd1, Sod1, and MP88) in *Escherichia coli* and loaded these antigens into GPs as a potential vaccine candidate [50]. Different mouse strains were vaccinated with these antigen-laden GPs and challenged with *C. neoformans* and *C. gattii*. The results showed varied protection depending upon the antigen, mouse strain, and cryptococcal species [50]. Furthermore, vaccination with GP containing *C. neoformans* Cda1 and Cda2 induced robust protective T_H_1 and T_H_17 responses. In these recent GP-Cda1 and GP-Cda2 vaccination studies, murine models deficient in pro-inflammatory cytokines IFNγ, IL-1β, IL-6, or IL-23 were not protected upon live *C. neoformans* challenge [51]. These studies emphasize the idea of employing cell-wall protein antigens as a novel vaccine adjuvant or delivery system against cryptococcosis.

## 4. Cryptococcal Chitin and Chitosan

Chitin is virtually present in all fungi; it is arranged into microfibrils to provide strength and rigidity. *C. neoformans* is chitin-rich and produces 3–6 times more chitin, which increases as the density of the culture progresses, as compared to the model yeast *S. cerevisiae*. Membrane protein chitin synthase (CHS) encodes eight CHSs in different classes based on the protein sequence of the catalytic domain [52]. The enzymes in classes I-III have seven transmembrane domains, while classes IV-VI have six predicted transmembrane domains [53]: Chs1 and Chs3 (class IV); Chs2 and Chs7 (class II); Chs4 and Chs5 (class V); and Chs6 and Chs8 (classes I and II). Chitosan, the deacetylated form of chitin, is formed through enzymatic conversion of N-acetylglucosamine to glucosamine by chitin deacetylases (CDAs). Three chitin deacetylases genes are essential for chitosan production in *C. neoformans*. Chitosan is mainly produced during the vegetative phase of growth and increases with culture density. The coordinated activities of Cda1 and Cda2 are essential for cryptococcal virulence [54]. Deletion of genes associated with chitin and chitosan result in an array of phenotypes, including temperature sensitivity, lack of chitosan, altered cell-wall integrity, budding defects, leaky melanin; and enlarged capsule thus rendering the cells avirulent [52,55].

Innate recognition of PAMPs induces a strong adaptive response [56]. Chitin is one of the PAMPs present in the cell wall that is required for virulence, and chitosan deficiency alters the Th1-based protective host responses [57,58,59]. Upadhya et al. emphasized how different culture media and pH values affect the amount of chitin and chitosan in the cell wall; in turn, these changes alter the cell-wall architecture and host response [60]. Vaccination with a chitosan-deficient *cda1Δ2Δ3Δ* (lacking all three chitin deacetylases) strain conferred protective immunity in mice to a subsequent challenge with a virulent wild-type *C. neoformans infection*. In contrast, mice infected with a chitosan-deficient *chs3Δ* strain died within 36 h post-infection due to an aberrant hyperinflammatory response, thus highlighting the critical immunomodulatory role of chitosan [61]. Vaccination with a single Cda protein induced cross-reactive antibody and IFN-γ immune responses to other Cda protein family members [62]. In summary, the complex structure of chitin is buried in the cell wall, shielded by mannoproteins and glucans. The strong elicitation of host protective immune responses by chitosan suggests that this is an attractive vaccine candidate with adjuvant and antigenic properties [56,63,64].

## 5. Mannoproteins

Mannoproteins are glycoproteins with heavily glycosylated mannose sidechains that have been identified as potential fungal antigens that can be targeted for vaccine development based on their ability to activate both innate and adaptive arms of the immune response. Their structure has been characterized as relatively conserved, having a signal peptide on the N terminus and a GPI anchor toward the C terminus with multiple N-linked and O-linked glycosylation sites throughout [65,66,67]. Mannoproteins either end up being secreted into the external environment or lodged in the cell wall due to their GPI anchors. Furthermore, the mannoprotein structure is characterized by having a Serine/Threonine-rich region for extensive O-linked mannosylation.

Mannoproteins have been reported to be involved in fungal virulence and cell-wall structural integrity in multiple fungal species [68,69,70]. For example, *C. albicans* mannoprotein MP-58 is located on the cell surface and has been found to elicit a strong IgG antibody response [68]. Monoclonal antibody treatment blocking the C terminal antibody-binding region of MP-58 in *C. albicans*-infected mice reduced mortality as compared to the non-treated cohort. The secreted mannoprotein Mp1p was shown to be a key virulence factor in the thermal dimorphic fungus *Talaromyces marneffei* [69,70]. Loss of Mp1p attenuated *T. marneffei* virulence seen and resulted in reduced survival and proliferation in macrophages. As previously mentioned, predicted *Cryptococcus* mannoprotein Krp1 was found to alter some virulence factors, including capsule thickness, cell-wall integrity, and phagocytosis in vitro in *C. gattii,* but had no effect in murine cryptococcosis models [19]. In studies focused on determining GPI anchor-containing mannoprotein MP-98, monoclonal antibodies against MP-98 were not able to bind to cryptococcal capsule in serotype A compared to serotype D, suggesting that mannoproteins are antigenically diverse throughout the serotypes of *Cryptococci* [18]. Most of the research on mannoproteins has focused on MP-84, MP-88, and MP-98. Over 40 potential mannoproteins in *C. neoformans* have yet to be characterized for function and contribution to cell physiology and virulence [65,71].

Fungal cell glycosylation differs from mammalian glycosylation in that mannose is used to extend the branched glycosylation sites. Mammalian cells use monosaccharaides and a multitude of glucotransferases to alter the N-glycan branching. A major difference between O-linked and N-linked glycosylation is the specificity of branching glycan sidechains added onto the amino acid Asparagine (N-linked) versus a linear addition of glycan sidechains to the hydroxyl group of Serine and Threonine amino acid-saturated regions of the protein.

The glycosylation of mannoproteins is a key contributor to their enhanced immunogenicity in *C. neoformans* [66]. Mansour and Levitz characterized the initial findings of mannoproteins as immunogenic antigens of *C. neoformans* that can trigger protective host immunity [72]. This and subsequent studies found that the mannoproteins MP-88 and MP-98 (Cda2) can stimulate T_H_1 protective cytokine production by CD4^+^ T cells [66,67,72]. Subsequent studies found that glycosylation of mannoproteins is key to the activation of dendritic cells triggered through danger-associated molecular pattern receptors like DC-SIGN that can subsequently induce T-cell response [71]. Additionally, MP-84 (Cda3) and MP-115 were identified as important targets of antibodies, as they react strongly with the sera of cryptococcal meningoencephalitis AIDS patients [18,73]. De-glycosylated recombinant versions of these MPs in *E. coli* induced a significantly weaker response in AIDS patients’ sera as compared to the naturally heavily glycosylated versions [73]. Furthermore, the differences in O-linked versus N-linked glycosylation have been identified to play an important role in the immunogenicity of cryptococcal mannoproteins. Recently, Su-Bin Lee and colleagues found that when core N-glycan structures were truncated in *C. neoformans* MP-84 (Cda3) and, to some extent, MP-98 (Cda2), the capacity to induce the immune response of bone marrow-derived dendritic cells was reduced [74]. Interestingly, complete ablation of N-glycosylation on MP-84 enhanced adhesion to host epithelial cells and increased cytokine production compared to the wildtype N-glycans [74]. This study emphasized the importance of the structure-dependent effects of N-glycans on the function of mannoproteins and lung-cell interactions. In aggregate, these findings highlight the importance of glycosylation in mannoprotein immunogenicity.

Cryptococcal vaccination studies involving MP-98 (Cda2) and MP-84 (Cda3) are a prime example of mannoproteins being identified as novel targets for fungal vaccination [51,60,75]. As previously referenced, Upadhya et al. showed that the triple chitin deacetylase-deficient strain *cda1Δ2Δ3Δ*, contains cell wall-integrity defects that contribute to its attenuated virulence and ability to induce protective cytokines in murine vaccination models [54]. Indeed, recent work aimed at developing multi-epitope subunit vaccines based on the chitin deacetylase (Cda1, Cda2, and Cda3) and MP-88 predicted that utilizing a combination of T-cell and B-cell epitopes together with adjuvants and linkers induced protective cytokine responses in silico [76]. While these findings would need to be validated and confirmed in both in vitro and in vivo models of *C. neoformans* infection, it highlights the novelty of “reverse vaccinology” utilizing an immunoinformatic approach to identify immunogenically favorable epitopes and test them in hypothetical models that reduce financial limitations and speed up vaccine-screening approaches. Furthermore, recent work by Wang et al. has shown that a quadrivalent cryptococcal subunit vaccine (Cda1+Cda2+Blp4+*cpd1Δ*) combined with Cationic Adjuvant Formulation 01 (CAF01) can induce a robust T_H_1 and T_H_17 CD4^+^ T-cell response for long-term protection against *Cryptococcus* [77]. Modulation of these cryptococcal cell antigens or other fungal species is critical to creating novel cryptococcal vaccines and for the identification of new anti-fungal drug targets. While the chitin deacetylase mannoproteins have been elucidated to be key functional mannoproteins that are themselves immunogenic, it emphasizes the need to further expound other mannoproteins in *Cryptococcus* pathogenesis that may hold the potential for vaccine use.

Cytokine-Inducing Glycoprotein 1 (Cig1) is another mannoprotein whose function is important for *Cryptococcus* survival in the host and holds immunogenic potential for vaccination studies. Cig1 was named for its ability to induce protective cytokines from immune cells and to interact with antibodies from serum obtained from AIDS patients with cryptococcosis [78]. Additionally, Cig1 has been found to be shed in media during *C. neoformans* growth. The mannoprotein Cig1 mediates iron uptake from the heme as a hemophore under iron-starved host-like conditions and contributes to virulence [79]. This impact on virulence is attenuated only when other proteins, including Cfo1, that allot functional redundancy in the iron uptake regulatory system of *C. neoformans* are also deleted. Like other mannoproteins, Cig1 contains a GPI anchor, as identified by Levitz and colleagues [65]. Cadieux et al. show that Cig1 is found to be excreted into the supernatant, as previously described, and that it is found toward the outside of the cryptococcal cell wall [79].

Interestingly, *CIG1* transcripts are highly abundant in *Cryptococcus* retrieved from the cerebral spinal fluid of cryptococcosis patients in clinical settings, further suggesting that Cig1 plays an important role in survival in harsh host conditions [80]. O’Meara et al. showed that the PKA pathway regulates pH-dependent transcription factor Rim101, which is known to regulate *CIG1* gene expression [10,81]. The protein kinase A (PKA) signaling pathway is also involved in regulating *CIG1* expression via Rim101. Further studies showed that capsule structure is altered where the *rim101Δ* mutant is hypo capsular yet propagates a hypervirulent phenotype [82]. Meanwhile, Geddes et al. showed that the cAMP/PKA pathway also regulates extracellular secretion of Cig1 in a PKA1-dependent manner observed in the *pka1Δ* mutant compared to the wildtype [83]. Follow-up studies from Geddes and colleagues observed a connection between the PKA pathway and the ubiquitin proteolysis pathway whereby the PKA expression alters proteostasis of virulence-related genes and endoplasmic reticulum control in capsule production [84]. Recently, the SCF (Skp1, Cullins, and F-box proteins) E3 ligase ubiquitin complex was identified to regulate Crk1, a CDK-related kinase in *C. neoformans* [85]. This study found that Crk1 is a substrate of F-box protein 1 (Fbp1) and a downstream regulator of the cAMP/PKA pathway via phosphorylation of Gpa1 [85]. In the *fbp1Δ* mutant, proper ubiquitin tagging and degradation are lost. This resulted in Crk1 accumulation and the induction of titan cell formation. Furthermore, overaccumulation of Crk1 attenuated virulence and increased induction of T_H_1/T_H_17 cytokines by CD4^+^ T cells. The connection between Cig1 and the cAMP/PKA pathway via Rim101, combined with recent findings identifying Crk1 as a novel regulator of cAMP/PKA, suggest there could be a potential role of Cig1 as an immunogen in the whole-cell heat-killed Fbp1-deficient (HK-*fbp1*) vaccine candidate. While studies with *cig1Δ* deletion mutants have focused on functionality in *C. neoformans* in iron uptake, the immunogenicity of Cig1 in the context of vaccination and ability to prime protective CD4^+^ T cells remain to be seen.

## 6. Extracellular Vesicles as Fungal Vaccine Platform

Extracellular vesicles (EVs) are small membrane-bound particles released from cells (mammalian, plant, and fungal) that play an important role in cellular communication [86,87,88,89]. Fungal EVs are known to participate in many biological processes, including transfer of virulence factors in *C. neoformans* to elicit a robust antigenic response in the host [89]. Cryptococcal EVs have been found to contain several membrane-bound protein families [90]. A recent study by Rizzo et al. suggested a new EV structural model where the vesicular bilayer is decorated with mannoprotein-based fibrils surrounded by capsule polysaccharide as its outer layer [90]. Furthermore, authors identified MP-88 and Vep1 on the surface, thus enabling its potential as a vaccine [90]. To test this idea, Rizzo et al. isolated EVs from WT and acapsular *cap59Δ* mutant strains and immunized BALB/c mice intraperitoneally. After three vaccine doses, mice were challenged with wildtype *C. neoformans*. EV-immunized mice survived longer than the non-immunized mice. Furthermore, EV *cap59Δ*-immunized mice showed prolonged survival compared to the WT EV-immunized mice [90]. A potential mechanism of conferred EV vaccine protection in this study is that EVs deliver the antigen directly to antigen-presenting cells (APCs) and are easily engulfed by APCs to elicit a cascade of protective immune activation.

Another study by Colombo et al. explored the use of *sgl1Δ* mutant EV as a vaccination strategy for cryptococcosis by utilizing an invertebrate model of the cryptococcal infection *Galleria mellonella* [91]. The *sgl1Δ* lacks sterylglucosidase, thus leading to the accumulation of sterylglucosides (SGs), acting as immunostimulatory glycolipids, inducing protection in murine model of cryptococcosis [91,92]. As EVs contain GXMs and SGs, using EVs as a vaccine strategy delayed acute lethality in *Galleria mellonella*. The possible reason for the success of *sgl1Δ* EV might be its composition and the larger size, resulting in increased interaction between host cell and EVs and inducing a strong immune response. Although *sgl1∆* protected the host from lethality in the beginning, all the *Galleria mellonella* later succumbed to death. While this was not shown in mammalian models of *Cryptococcus* infection, it highlights that EVs might boost the host immune system, even though they fail to offer complete protection. Future studies should explore new vaccine strategies implementing EVs as a vaccine adjunct to boost host immune response. Overall, the study highlighted potential use of Sgl1 and its enriched EVs as a cell-free vaccine formulation that would be a particularly attractive vaccine candidate.

In conclusion, the advantages of EV-based vaccines include the versatility with which this strategy can be modified to carry multiple antigens and the fact that such vaccines are relatively biocompatible and have minimal toxicity. The potential challenges in EV-based vaccines could be the cost, since the process of producing EVs and purifying is complex and time-consuming compared to traditional vaccine platforms. Furthermore, depending on the origin, some EVs might have an immunosuppressive effect, thus affecting vaccine efficacy. Fungal EVs offer a promising vaccine candidate and innovative approach to deliver antigens effectively and stimulate robust immune responses. Future studies may focus on engineering the EVs by modifying their contents and surface markers to boost host protective responses and overcome vaccine delivery challenges.

## 7. Immunological Responses of Current Whole-Cell Cryptococcal Vaccine Candidates

Immunocompromised populations lacking either the adaptive or innate arm of the host immune response are highly susceptible to an array of fungal species (i.e., *Cryptococcus*, *Aspergillus*, *Candida*, *Coccidioides*, *Histoplasma*, and *Blastomyces*). This susceptibility will continue to rise with expanded use of immunosuppressant treatments to remedy other diseases. One of the defining risk factors for *Cryptococcus* infection is the lack of CD4^+^ T-cell populations, as observed in the high percentage (~19%) of *Cryptococcus*-related deaths in HIV/AIDS patients [4]. Thus, an ideal vaccine candidate needs to be able to circumnavigate the loss of this immune cell population and be able to induce long-term protection [93]. Exciting developments in understanding both the innate and adaptive immunological mechanisms of protection in response to several whole cell-based vaccine candidates has greatly contributed to identifying how we can manipulate cryptococcal antigens to induce desired immune responses in vaccine development.

Currently reported whole cell-based cryptococcal vaccine candidates include a human interferon γ-producing genetically modified strain (H99γ) [94,95,96,97]; a sterylglucoside-deficient strain (*sgl1Δ*); an overexpressed mating specific zinc finger transcription factor strain that restrains cryptococcal cells in a pseudo-hyphal morphological stage (Znf2^OE^) [26,98,99]; an F-box protein deletion strain that alters SCF E3 ligase protein proteolysis pathway (*fbp1Δ*) [100,101,102,103]; and the chitin deacetylase triple mutant (*cda1Δcda2Δcda3Δ*) [54,58,59,61,75,104] (Table 1). Each of these strains has shown successful protection against *C. neoformans* challenge following vaccination in different mouse-strain backgrounds. Furthermore, these strains appear to share somewhat conserved immunological mechanisms of action in protective anti-fungal responses.

## 8. Adaptive Immunity

The adaptive immune response plays a critical role in host defense against *Cryptococcus* species. Protection against *C. neoformans* infection is primarily mediated via IFNγ and IL-17A cytokine-producing CD4^+^ T cells’ responses in immunocompetent settings [107]. Expansion of these CD4^+^ T-cell populations while balancing the influx of T_H_1 and T_H_17 cytokines responses, is also important for the host survival following vaccination and subsequent challenge. Importantly, the production of these cytokines must still be preserved in CD4^+^ T cell-deficient settings in vaccination applications for *C. neoformans*. Protection against *C. neoformans* is challenged by these T_H_1 and T_H_17 responses, which are similarly induced by all current cryptococcal vaccine candidates highlighted in this review (Figure 2). The live-attenuated H99γ strain has been shown to confer protection in CD4^+^ and CD8^+^ T cell-depletion models, but protection is lost when the mutant is in heat-killed form [94,108,109]. However, many of the listed candidates can continue to confer protection in heat-killed forms in CD4^+^-deficient host models so long as other T-cell compartments are intact (Figure 2). For instance, the *sgl1Δ* vaccine can confer protection in either CD4^+^ or CD8^+^-deficient backgrounds, so long as one or the other is intact. In double CD4^+^/CD8^+^ T cell antibody-depletion murine models, *sgl1Δ* vaccine protection is lost. and mice succumb to infection following H99 challenge [110]. CD4^+^ T cells were found to be required for vaccine-mediated protection against other cryptococcal strains, including *C. gattii*, but it is unknown whether this vaccination can provide cross-protection against other fungal pathogens. Interestingly, protection conferred by *sgl1Δ* vaccination is dependent on γδ T cells that can produce key protective cytokines, IFNγ and IL-17A [27]. In the HK-*fbp1* vaccine candidate, RAG^-/-^ mice genetically lacking the entire adaptive immune response (CD4^+^, CD8^+^, and B cells) also succumb to infection following challenge [100,103]. Wang and colleagues identified that when the CD4^+^ T-cell compartment was depleted in the HK-*fbp1* vaccination model, the CD8^+^ T-cell compartment would expand and produce key protective cytokines to compensate for the other population’s loss [100]. Furthermore, the HK-*fbp1* vaccine is the only cryptococcal vaccine candidate that has been shown to provide cross-protection against other fungi, such as *Aspergillus* and *Candida.* This protection against *A. fumigatus* infection was also surprisingly conferred in a neutropenic murine model. In the context of *Aspergillus* infection, where monocytes and neutrophils are critical innate immune cells that orchestrate anti-fungal immune responses [111,112,113], it was exciting to see that vaccination with HK-*fbp1 C. neoformans* was able to provide heterologous protection against other fungal species [100]. In summary, these findings emphasize that while IFNγ and IL-17A production by both CD4^+^ or CD8^+^ T cells can confer protection in these models, there may be a role for other sources of these protective cytokines (i.e., γδ T cells) in some vaccine candidates. Furthermore, vaccination with these strains may be able to induce cross-protection against other medically relevant fungal pathogens.

The role of B cell-mediated immunity in defense against *Cryptococcus* infection has been found to have a moderate impact in both immunocompetent and compromised settings. Moreover, the impact of antibody-mediated protection has been found to vary based on *Cryptococcus* strain and murine models used to investigate humoral responses [114]. B cell-mediated immunity was dispensable for protection in live H99γ-vaccinated mice since B cell-deficient mice survived H99 challenge [109]. B cells are also found to be dispensable in the *cda1∆cda2∆cda3∆* vaccination [115]. In studies on the impact of humoral response in *sgl1Δ,* vaccination models employed CD19-depleted murine models and found a minimal impact for B cells in this model [110]. In the Znf2^OE^ vaccination model, IgG and IgM antibodies from Znf2^OE^ vaccinated mice showed higher-intensity binding to Znf2^OE^ *C. neoformans* cells compared to the H99 strain, indicating that antibody titers of vaccinated host are increased [26]. However, the role of B cells in the cryptococcal vaccination candidates Znf2^OE^ and *fbp1Δ* have not been explicitly investigated in either live or heat-killed vaccination forms.

## 9. Innate Immunity and Trained Innate Immunity Responses

While the adaptive response is key to preventing dissemination in *Cryptococcus* infection, the role of the innate immune system cannot be underestimated. The innate immune arm comprising first-responder phagocytes (macrophages, monocytes, and neutrophils) and professional antigen presenting dendritic cells are required for priming CD4^+^ T cell responses. Specifically, C-type lectin receptors (i.e., Dectin-1, Dectin-2, DC-SIGN, and Mincle) and Toll-like receptor (TLR2, TLR4, and TLR9) are established pattern-recognition receptors (PRRs) that play key roles in recognition of fungal antigens (i.e., mannans and β-glucans) that are required for priming and expanding T_H_1 and T_H_17 CD4^+^ T cells [116,117,118,119,120,121]. Furthermore, this priming of effector CD4^+^ and activation of cytolytic CD8^+^ T cells is required in vaccine-mediated protection in multiple models for combating infectious disease [122]. Excitingly, the role of “trained immunity”, whereby innate immune cells have the capacity to retain immunological memory, originally thought to be retained only by T and B lymphocytes, has been shown to play a role in anti-fungal vaccination [123,124,125]. For example, trained innate immunity of monocytes in transition to macrophages undergo STAT1-dependent epigenetic reprogramming driven by H99γ strain IFNγ production [95] (Figure 3). In this model, protection continued up to 70 days post-H99 challenge in the absence of B cells, CD4^+^ T cells, neutrophils, and NK cells [95]. These findings emphasized the impact of trained innate immune memory against specific cryptococcal antigen independent of canonical antigen presentation by APCs to CD4^+^ T cells. However, these methods of protection are apparent only in live-attenuated versions of this strain, and protection is lost in the heat-killed format. In recent studies, Wang and colleagues identified that monocytes and neutrophils are important producers of IFNγ after HK-*fbp1* vaccination [102]. Furthermore, STAT1 expression in CD11c^+^ cells (alveolar macrophages, monocyte derived macrophages, and monocyte derived dendritic cells) is required for vaccine-induced protection [102]. Currently, other whole-cell *C. neoformans*-based vaccine candidates have shown correlations of increased inflammatory cytokine production in the lung milieu, along with increased leukocytes following vaccination and subsequent live challenge [51,54,60,61,110]. However, evidence of epigenetic reprogramming of leukocytes attributed to innate immunity training in the remaining vaccine candidates has yet to be elucidated. In aggregate, current studies from several groups suggest that trained innate immunity plays a critical role in the cryptococcal vaccine candidate that future studies need to focus on as research pivots eventually to multi-valent subunit vaccine approaches.

## 10. Concluding Remarks

*Cryptococcus neoformans* is a serious fungal pathogen that continues to be a growing global health concern. Over the last 40 years, the mycology field has made significant strides in elucidating the complexity of the cryptococcal cell wall and the antigens that induce protective host immunity. While multiple studies have worked to identify single immunogenic antigens that can be modulated or overexpressed for fungal vaccination candidates, it is unlikely that a single antigen-based vaccine will be successful. The synergistic use of multiple fungal components—glucan, chitin, and chitosan—will most likely lead to effective host protective responses. Fungal vaccine research is moving toward the development of multivalent peptide or subunit-based vaccines. Utilizing this approach would broaden protection by targeting multiple epitopes across fungal species and reduce safety concerns in comparison to attenuated or heat-killed whole-cell vaccines that may elicit unintended immune activation against cross-reactive host antigens. Excitingly, computational analysis and in silico approaches have been used to identify immunodominant MHC I and II and B cell epitopes against *Coccidioides* and *Candida albicans* antigens, leading to viable multivalent peptide-based vaccine candidates [126,127].

With the increasing number of cryptococcal meningitis cases coupled with anti-fungal drug resistance, we need to translate our in vivo murine findings to human trials, as well as investigate adjuvants to enhance the immunogenicity of current fungal vaccine candidates. Overall, the versatility of fungal cell components in drug delivery and the ability to induce immunogenic response in combination therapies, can offer broader therapeutic anti-fungal applications. Indeed, we have seen progress in the recent novel drug delivery systems, such as DectiSomes, comprising Amphotericin B-loaded pegylated liposomes coated in Dectin-3 polypeptide. This drug delivery tool was first demonstrated by Chaudury and colleagues to bring Amphotericin B directly to fungal pathogens, including *Candida albicans*, *Rhizopus delemar*, and *Cryptococcus neoformans*, by utilizing the specificity of C-type lectin receptors to yeast α-mannans [128]. DectiSomes were later modified with Dectin-2 to effectively target cryptococcal cells in vivo following challenge [129]. The development of such integrative approaches, in combination with harnessing immunogenic cryptococcal antigens that elicit protective immune responses, is the next step forward in attaining a safe and effective cryptococcal vaccine for high-risk immunocompromised patients.

## Figures and Tables

**Figure 1 jof-10-00840-f001:**
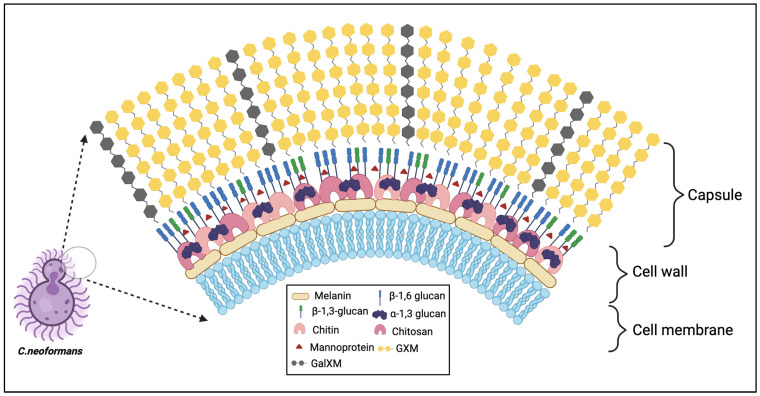
Structure and composition of *C. neoformans* capsule and cell wall. Antigenic factors that influence the host immune response and comprise the *C. neoformans* cell wall include melanin, chitin, chitosan, α and β-glucans, and mannoproteins. The capsule comprises GXM and GalXM.

**Figure 2 jof-10-00840-f002:**
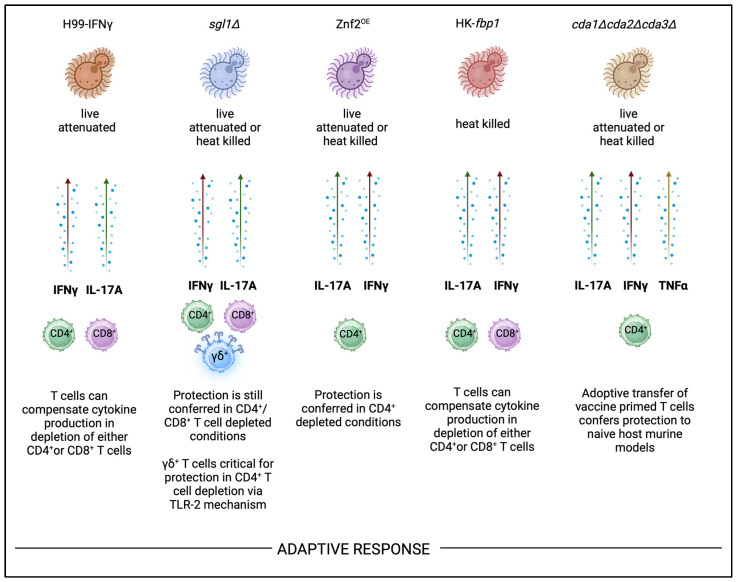
Adaptive immune response to attenuated whole-cell cryptococcal vaccine candidates.

**Figure 3 jof-10-00840-f003:**
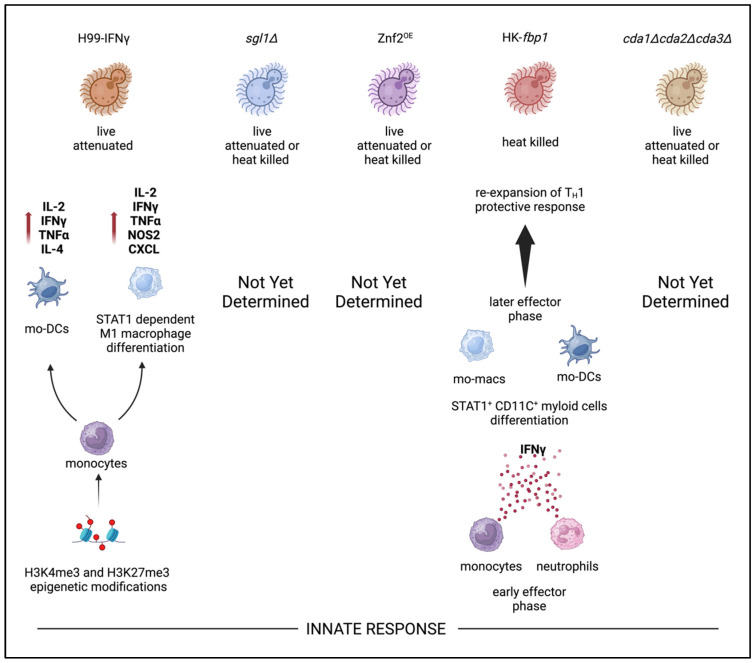
Innate immune response to attenuated whole-cell cryptococcal vaccine candidates.

**Table 1 jof-10-00840-t001:** Cryptococcal vaccine candidates and their mechanism of action.

Vaccine Candidate	Vaccination Method	Background	Vaccine Route Administration	Mechanism	Reference
*sgl1Δ*	Whole-cell, live-attenuated, and heat-killed	*C. neoformans* sterylglucosidase deficient strain	Intranasal	IFNγ and IL-17A produced by γδT CD4^+^, and CD8^+^cells	[27]
H99γ	Whole-cell, live-attenuated	Mouse IFNγ producing *C. neoformans* H99 strain	Intranasal	T_H_-1/proinflammatory cell response	[94]
Znf2^OE^	Whole-cell, live-attenuated, and heat-killed	*C. neoformans* zinc finger transcription factor 2 overexpressed	Intranasal	T_H_-1/T_H_-17	[97]
HK-*fbp*1	Whole-cell, heat-killed	Disruption of SCF E3 ligase complex via deletion of F-box protein 1 in *C. neoformans*	Intranasal	T_H_-1/T_H_-17 response	[100,103]
*cda1* *∆2* *∆3* *∆*	Whole-cell, live-attenuated, and heat-killed	Deletion of 3 chitin deacetylases in *C. neoformans*	Intranasal	CD4^+^ T-cell response; proinflammatory cytokines IL-1β, IL-6, and IL-23	[54]
Glucan Particles (GP)	Protein subunit vaccine	Synthesized subunit protein	Intranasal	Antibody and T-cell response	[48,49,51]
β-Glucan antibody	Antibody-based	Monoclonal antibody	Intraperitoneal	Antibody response	[105]
Glucosylceramide antibody	Antibody-based	Monoclonal antibody	Intraperitoneal	Antibody response	[106]
P13-TT	Antibody-based	Peptide mimic of *C. neoformans* GXM conjugated to tetanus toxoid	Subcutaneous	Antibody response	[34]
GXM-TT	Subunit vaccine	*C. neoformans* GXM conjugated to tetanus toxoid	Subcutaneous	Antibody response	[34]
GalXM-BSA	Subunit vaccine	*C. neoformans* GalXM conjugated to BSA	Subcutaneous	Antibody response	[36]
GXM antibody 18B7	Antibody-based	Monoclonal antibody	Intravenous	Antibody responseclinical trial phase 1	[31]

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
