# Peer review of "Will the Real Immunogens Please Stand Up: Exploiting the Immunogenic Potential of Cryptococcal Cell Antigens in Fungal Vaccine Development"

_jof, 2024, doi:10.3390/jof10120840_

Round 1
Reviewer 1 Report
This article with 129 references reviews research on the development of vaccines against the fungi Cryptococcus. The authors have made a lot of efforts to summarize the work presented in the references and the review has its merits but also contains, in my opinion, major flaws and mistakes (see below) why I can’t recommend publication.
Some flaws and mistakes:
In the abstract it says, “whole cell-based and protein subunit-based vaccines”, but many of the subunit used are carbohydrates.
In Figure 1 it seems that the amount of GXM and GalXM is the same, while it’s actually GXM/GalXM 9:1 (as said in the text).
On line 136-137 it says that “Other GXM-based vaccine approaches have focused on conjugating a GXM mimicking peptide, P13, to immunogenic antigens such as tetanus toxin (P13-TT)” with references 32-34 cited, but in references 32 and 33 native GXM (conjugated to TT) is used not the peptide mimic. Also, the Cryptococcus-related antigen in these experiments is the GXM (native or mimic), the TT is used as a carrier protein to produce a T-cell dependent response (towards the GXM antigen).
Table 1: What does the “recombinant vaccination method” listed for the GXM-TT and GalXM-BSA Vaccine Candidates mean? As mentioned above the relevant immunogen is the GXM and GalXM and neither the TT nor the BSA used as carrier protein is recombinant.
Table 1: How can there be a column titled “Mutant background” when the immunogen is a carbohydrate”?
Table 1: It says that all vaccine route administrations are intranasal, I think all of them are through injections.
None
Reviewer 2 Report
The manuscript by Avina et al. reviewed the immunogenic potential of Cryptococcal cell antigens, which is a scientific topic crucial to the development of a promising fungal vaccine against Cryptococcosis. The authors’ discussion covered almost all well-known cryptococcal cell immunogens, including polysaccharide capsule, glucans, chitin/chitosan, mannoproteins, and extracellular vesicles. The topic was thoroughly discussed by introducing the role of each antigen in the structural composition and virulence of Cryptococcus, outlining the immunogenicity of the antigens and possible mechanisms of host immune response related, and most importantly, comprehensively summarizing current evidence or future possibilities of their role in novel cryptococcal vaccination approaches. Overall, the manuscript is structurally well organized and well written. The viewpoints discussed are well supported with updated references. A few issues are as below in the detail comments.
1. The ratio of GXM vs. GalXM is around 90%:10%, and they are interconnected polymers that decreases in density, porosity when extending radially outward from the cell wall. The proportion of GXM and GalXM, and the way they connected and layered in Figure 1 seemed not to be demonstrating this point appropriately. The figure illustrated both cell wall and capsule. The figure legend could be changed to reflect the content.
2. Line 131, typing error “monoclonal anti-)”, should be monoclonal antibody 18B7.
3. Line 147-150 The content is about GXM based vaccine as well, it will be better to remove to the end of line 141, before GalXM.
4. Line 207, in the literature referenced, a robust IFNγ/IL17/TNFα response from CD4+ T cells are detected. The loss of vaccination protection in mice with genetic deficiency in IL-1β, IL-6 and IL-23 suggested these pathways could be important for the vaccine induced immunity as well. However, the response for these three cytokines were not directly tested in this study.
5. Line 423 Table 1, Please double check the vaccine administration route. Some of the vaccines like GP vaccines are administered subcutaneously rather than intranasal. And it’s better to keep it consistent in the context and table for the name of deletion strains. It’s great to summarize other vaccine candidates as well but not just whole cell vaccines in the table, however, the column title “Mutant Background” does not apply to the non-whole cell vaccine condition.
6. In Fig 2, for live cda1Δcda2Δcda3Δ vaccines, research data showed the protection was lost in mice genetically deficient in the cytokine IL-23, suggesting its crucial role. The IL-23 is essential for the differentiation and maintenance of Th17 cells. However, it is mainly produced by dendritic cells and macrophage towards stimulation, binding with IL-23 receptors expressed on CD4 T cells, NK cells and other cells. It’s not quite appropriate to list it as a cytokine produced by CD4 T cells in the figure.
7. Line 507, typing error “(ref)”. The reference is missing.
